# Natural Products in Renal-Associated Drug Discovery

**DOI:** 10.3390/antiox12081599

**Published:** 2023-08-11

**Authors:** Wasco Wruck, Afua Kobi Ampem Genfi, James Adjaye

**Affiliations:** 1Institute for Stem Cell Research and Regenerative Medicine, Medical Faculty, Heinrich Heine University, Moorenstr. 5, 40225 Düsseldorf, Germany; wasco.wruck@med.uni-duesseldorf.de; 2Department of Biochemistry and Molecular Biology, Faculty of Biosciences, University for Development Studies, Nyankpala P.O. Box TL 1882, Ghana; agenfi@uds.edu.gh; 3EGA Institute for Women’s Health, Zayed Centre for Research into Rare Diseases in Children (ZCR), University College London (UCL), 20 Guilford Street, London WC1N 1DZ, UK

**Keywords:** antioxidant, anti-inflammatory, AKI, CKD, natural medicinal products

## Abstract

The global increase in the incidence of kidney failure constitutes a major public health problem. Kidney disease is classified into acute and chronic: acute kidney injury (AKI) is associated with an abrupt decline in kidney function and chronic kidney disease (CKD) with chronic renal failure for more than three months. Although both kidney syndromes are multifactorial, inflammation and oxidative stress play major roles in the diversity of processes leading to these kidney malfunctions. Here, we reviewed various publications on medicinal plants with antioxidant and anti-inflammatory properties with the potential to treat and manage kidney-associated diseases in rodent models. Additionally, we conducted a meta-analysis to identify gene signatures and associated biological processes perturbed in human and mouse cells treated with antioxidants such as epigallocatechin gallate (EGCG), the active ingredient in green tea, and the mushroom Ganoderma lucidum (GL) and in kidney disease rodent models. We identified EGCG- and GL-regulated gene signatures linked to metabolism; inflammation (NRG1, E2F1, NFKB1 and JUN); ion signalling; transport; renal processes (SLC12A1 and LOX) and VEGF, ERBB and BDNF signalling. Medicinal plant extracts are proving to be effective for the prevention, management and treatment of kidney-associated diseases; however, more detailed characterisations of their targets are needed to enable more trust in their application in the management of kidney-associated diseases.

## 1. Introduction

Chronic kidney disease (CKD) and acute kidney injury (AKI) have emerged as major public health burdens with close connections to each other, AKI as a risk factor for CKD and vice versa and both increasing the risk of cardiovascular disease [1]. CKD is defined by a low glomerular filtration rate (GFR) or the presence of kidney damage for more than 3 months [2,3]. Proteins in urine (proteinuria) and decreased GFR as indicators of kidney damage directly reflect the physical properties of the filter between blood and urine constituted by an endothelial layer, the glomerular basement membrane (GMB) and podocytes [4]. While, under physiological conditions, most proteins cannot traverse this barrier, in proteinuria, larger proteins such as albumin, immunoglobulins G and M and α1-microglobulin and β2-microglobulin, correlating with the severity of the histologic lesions [5], can. These proteins can, as a consequence, impair the reabsorption of other, smaller molecules by proximal tubular epithelial cells and ultimately lead to toxic damage [5]. As risk factors of CKD, there has been a global rise in incidences of diabetes and hypertension [6,7].

AKI is defined by a rapid increase in serum creatinine concentrations and/or decline of urine output. The number of incidences is approximately 10–15% of patients admitted to hospital and approximately more than 50% in intensive care units [8,9]. Distinct time intervals of endurance of the pathological conditions are used to distinguish between AKI (<7 days), acute kidney disease (AKD; 7 days–90 days) and CKD (>90 days) [9]. AKI is now regarded as a multiorgan dysfunctional disease and classified as prerenal AKI, acute postrenal obstructive nephropathy and intrinsic acute kidney disease, of which only the latter is a true renal disease [10,11].

## 2. Plant-Based Extracts with Antioxidant and Anti-Inflammatory Properties

Natural products have, for centuries, been used in the management of various disease. Mother nature has served and is still serving us well. Plant-based extracts serve as natural sources of antioxidant and anti-inflammatory agents in combating stress, inflammation and cell death. Numerous oxygen-based metabolic activities generate reactive oxygen species, which can serve as signalling molecules. These signalling molecules serve as precursors of various beneficial events for the body. Increased levels of reactive oxygen species (ROS) and/or reactive nitrogen species (RNS) oxidative stress are an imbalance in the levels of ROS and the body’s natural antioxidant capacity. This creates complications due to ROS reacting with membranes and biomolecules such as lipids and proteins, thus leading to organ damage.

Oxidative stress and inflammation are known to cause various diseases, hence the need to have antioxidants and anti-inflammatory agents that will serve in combating oxidative stress-associated diseases. Increased levels of ROS and RNS are known as potential inducers of kidney injury [12,13,14,15], and molecules associated with ROS and RNS are major regulators of solute and water reabsorption in the kidney [16].

In the assessing and diagnosis of AKI, interleukin-6 (IL-6), interleukin-1 (IL-1), tumour necrosis factor (TNF), adipokines, adhesion molecules and the CD40 ligand are proinflammatory cytokines, which are indicative of the extent of stress or inflammation [17,18,19,20]. Ferguson et al. 2008 [21], Shinke et al. 2015 [22] and Zhou et al. 2008 [23] have also stated the importance of kidney injury molecule-1 (KIM-1) and neutrophil gelatinase-associated lipocalin (NGAL) as additional urinary biomarkers.

Medicinal plants naturally have antioxidant systems that help in combating oxidative stress. Superoxide dismutase (SOD), catalase, glutathione (GSH), which helps in drug metabolism detoxification, and glutathione peroxidase (GPx) are the systems that help in combating oxidative stress. Each medicinal plant extract has its own antioxidant-based mechanism for managing oxidative stress. Some increase the levels of SOD [24] and activate the SOD and catalase levels [25,26], while some plants increase the levels and activities of all the three antioxidant enzymes: SOD, catalase and GPx [27,28].

Natural medicinal plants have showed in vivo and in vitro efficacy in downregulating proinflammatory cytokines and upregulating natural antioxidants such as glutathione, which modulates oxidative stress and inflammation. Though modern treatment approaches have also afforded substantial progress in the fight against AKI, potent therapies are still meagre due to a lack of oxidative stress- and inflammation-specific AKI targets. The cost effects coupled with harsh side effects approaches have led to the search for more novel natural biological products, especially those derived from herbs and natural spices. These can all be accessed in medicinal plants due to the various Phyto components present in them. Their uses and applications are promising, since they are able to react, bind, conjugate and possibly eliminate through excretion all reactive oxygen spices (ROS and RNS), which form the bases of most ailments by cell degradation, lipid peroxidation and inflammation, among others.

## 3. Plant Sources and Activity

Plants have various phytochemicals with the potential to combat ailments. Research on medicinal plants is being carried out mostly in Asia, followed by Africa and Europe. Below are plants, herbs and spices that have been scientifically proven to manage, protect or cure acute kidney injury.

*Aronia melanocarpa* of the black chokeberry family mostly found in North America contains anthocyanins (cyanidins), which are able to potentially decrease inflammation, oxidative stress and lipid peroxidation, as well as apoptosis, in acute renal ischaemia effectively [29].

Kang et al. 2021 [30] showed that green tea is rich in bioactive compounds. Its aqueous high content of antioxidants has made it active in managing oxidative stress, resulting in the development of various healthy and nutritious detoxification products.

*Punica granatum* is a plant originally from India. Administration of the fruit peel ethanolic extract in Wistar rats showed improvement in kidney function biomarkers, exerted antioxidant activity and ameliorated histological changes prerenal and intrinsic gentamicin-induced AKI [31,32].

The methanolic peel extract of passion fruit (*Passiflora* spp.), which is predominantly found in North America, contains gallic acid, ellagic acid, kaempferol and quercetin glycosides. The extract is able to protect the kidneys by maintaining the levels of urea and creatinine at normal units during paracetamol-induced nephrotoxicity in albino rats [33]. The methanolic extract of its upper parts reduced the urea and creatinine levels during thioacetamide-induced nephrotoxicity in Sprague–Dawley rats [34].

*Pistacia atlantica*, an exotic berry-like fruit plant, is mostly be found in North Africa, the Middle East, Iran and Afghanistan. Leaf hydroethanolic extracts of *Pistacia atlantica* have the ability to decrease the levels of urea, creatinine and uric acid during gentamicin-inducted nephrotoxicity in Wistar rats [35].

*Eurycoma longifolia* is an herbal medicinal plant mostly found in Southeast Asia, Indonesia. The standardised aqueous extract of the roots has been shown to increase the levels and activities of antioxidant enzymes and improves kidney function during paracetamol-induced nephrotoxicity in rats [36].

*Costus afer* is an African indigenous plant used traditionally for the treatment of several diseases, such as rheumatoid arthritis, hepatic diseases, measles and malaria, and can also serve as an antidote for snake poisoning [37]. The aqueous extract of the leaf has been shown to decrease the serum potassium and BUN levels [38]. It also uses its antioxidant and anti-inflammatory potential to provide neuroprotection against low-dose heavy metal mixed neurotoxicity [39].

*Ocimum americanum* (family Lamiaceae) grows in Africa, India, China and Southeast Asia and is used as a spice. In Ghana, it is widely cultivated (called akokobesa) [40] and also used by locals to manage diabetes [41]. Nyarko et al. reported that it reduces blood glucose in mice and improved insulin release in beta cells isolated from rats [41]. Genfi et al. reported a hepatoprotective effect of *Ocimum americanum*, probably due to the inhibition of oxidative stress and the downregulation of proinflammatory cytokines [40].

Cranberry (*Vaccinium* sp.) natural extracts from North America decrease E. coli adhesion and reduce bacterial motility and biofilm formation in urinary tract infections [42]. Its polyphenols have anti-inflammatory and antioxidant effects and also have positive effects on the gut flora [43].

*Descurainia sophia* is a dominant weed with several local names and mostly found in Europe and Northern Africa. Csikós et al. 2021 [44] studied its effect on Wistar rats; its aqueous seed extract decreases the deposition of calcium oxalate in ammonium chloride and ethylene glycol-induced gallbladder stones.

An extract of the aerial parts *Equisetum arvense*, a fern-like plant mostly found in Spain, heals urinary retention and urinary infections, among others [45,46].

The aqueous leaf extract of *Anchomanes difformis* decreases the levels of oxidative stress-associated biomarkers and increases the CAT and SOD levels in African Wistar rats. It has anti-inflammatory effects by reducing the expression of NF-κB and Bcl2 and decreasing the levels of IL-10, IL-18 and TNF [47].

*Hibiscus sabdariffa* is a plant used for indigenous beverages in most parts of Asia, Africa and Central America. The aqueous extract of the dried flower bulb contains anthocyanins and chlorogenic acid, which increase both the enzymatic and nonenzymatic antioxidant systems [48].

*Curcuma longa*, a rhizome, is found mostly in India but now has been planted in Ghana. It contains polyphenol and is used for antioxidant, anti-inflammatory, antimicrobial and antitumour activity, among others [49].

The Lamiaceae family *Melissa officinalis* (lemon balm) is a well-known herb indigenously used to cure a variety of ailments [50]. It has glycosides that give it antioxidant and cytotoxic properties [51].

Mostly found across Europe is *Digitalis purpurea* L., a member of the Scrophulariaceae family [52]. The glycosides of D. purpurea have antioxidant and cytotoxic properties. Lycopene, β-carotene and the vitamins of tomato fruits also help to reduce oxidative stress and reduce the risk of cancer [53,54]. Oxidative regulation is paramount in the management of AKI. The aerial parts of Tylophora indica contain alkaloid and tylophorine, which serve as anti-inflammatory and immunosuppressive agents [55]. Lavandula intermedia leaves and flowers have been shown to contain polyphenols, which are significant in providing UV protection [56].

## 4. Active Ingredients

These plants, herbs and spices serve as antioxidants, anti-inflammatories, anti-malarias, anti-hyperglycaemias and hepatic protectants, among others. This is possible due to the active ingredients in them: flavonoids, alkaloids, saponins, tannins, coumarins, cyanides, anthocyanidins, phenols, phenolics, carotenoids, phytoestrogens [57,58], capsaicin [59,60], curcumin [61], β-carotene [62,63], catechins [64,65], resveratrol [66], vitamins, flavonoids (hyperoxide) and xanthones, as well as naphthodianthrone hypericin (antiviral action), the phloroglucinol derivative hyperforin (antibacterial effect) [67], cardiac glycosides [68], flavonoids, anthraquinones and triterpenes [69,70,71,72].

## 5. Meta-Analysis of Transcriptome Datasets Related to Natural Products

Table 1 lists datasets found by searches in the public database NCBI GEO (National Center for Biotechnology Information Gene Expression Omnibus). We excluded cancer-related datasets from the results, as we were more interested in the earlier inflammatory processes that may be transferred to AKI and CKD. We found three datasets associated with the same compound, EGCG (epigallocatechin gallate), the active ingredient of green tea, supplemented in human skin and T cells and mouse colon-derived cells. Taking into account the variability of the cell types, our strategy was to identify a gene signature common to skin and T cells. This could then be further refined by identifying conserved genes from mice in the EGCG-treated mouse colon dataset. In the final step, we aimed to transfer this signature to the kidneys by checking overlaps with the kidney datasets, which were made with different compounds but at least with antioxidant properties.

In the EGCG-treated transcriptome datasets (GSE152781) associated with the publication by Ud-Din et al. [73], GSE53448 associated with the publication by Kehrmann et al. [74] and GSE41644 associated with the publication by Barnett et al. [75], we looked for genes up- and downregulated with *p* < 0.05 (test of the R package limma [76]) and a positive or negative (downregulation) logarithmic fold change when comparing the EGCG treated with the control condition.

### 5.1. Comparison and Functional Annotation of Differentially Regulated Genes

Differentially up- and downregulated genes were compared in Venn diagrams via the R package VennDiagram [77]. Subsets of the Venn diagrams were analysed for overrepresented gene ontologies (GOs) via the R package GOstats [78]. The most significant GO terms were plotted in a dot plot via the R package ggplot2 [79]. A further enrichment analysis was performed via the R package EnrichR [80].

### 5.2. Construction of a Gene Regulatory Network

We looked for enrichment of the EGCG down- and upregulated genes with respect to the transcription factor dataset TRANSFAC_and_JASPAR_PWMs in the EnrichR database [80]. The transcription factor enriched for most EGCG-dysregulated genes were subjected to a protein interaction network analysis via STRING-DB [81]. Within the STRING-DB online tool, the network of the uploaded proteins was expanded in one step by five proteins, and the resulting network was analysed for overrepresented Reactome and WikiPathway pathways.

### 5.3. EGCG Upregulated Gene Signature

The Venn diagram in Figure 1A shows 23 (11 + 12) genes, which were identified as upregulated by EGCG in common in the skin and T cells. Further intersection with EGCG-treated mouse colon cells identified the 11 genes in the centre of the Venn diagram, which were thus supposed to have a conserved response to EGCG. Figure 1B,C show the most significant GO terms overrepresented in the 11 conserved genes and the 23 (11 + 12) human gene signatures. The large overlap between them indicates that most of these biological processes are conserved between men and mice. The significant GO terms can be classified into metabolic (amine and phenol metabolism), ion transport, membrane and secretory categories.

### 5.4. Gene Signature Downregulated by EGCG

The Venn diagram in Figure 2 shows that 50 (13 + 37) genes were found downregulated by EGCG in common in the skin and in T cells and, with further comparisons with EGCG-treated mouse colons, unveiled 13 conserved genes. The large overlap between both GO analysis results shows that most of these biological processes are conserved between men and mice. The results can be grouped into calcium signalling, neuronal and synaptic, inflammatory (response to wounding), connective tissue, ion transport and taxis categories. With respect to inflammatory processes in kidney diseases, the GO term “response to wounding” found significantly overrepresented in the downregulated EGCG signature (Figure 2B) was relevant. We compared this result with transcriptome data (GSE186823 and GSE171240) from two previous publications in which we investigated kidney organoids treated with the nephrotoxin puromycin aminonucleoside (PAN) [82] and urine-derived podocytes stimulated with angiotensin II (ANG II) [83]. The GO term “response to wounding” was also found significantly overrepresented in the genes downregulated in urine-derived podocytes upon ANGII treatment (*p* = 0.0015) and kidney organoids upon PAN treatment (*p* = 0.0354). Among the genes associated with “response to wounding”, we identified *NRG1* as overlapping in the downregulated EGCG signature (Figure 2B) and also in the downregulated genes upon PAN treatment. In the corresponding comparison with wounding-related GO terms in the downregulated EGCG signature, no overlapping genes were identified in the ANGII-stimulated podocytes.

### 5.5. Refined Characterisation and Transcription Factor Analysis of the EGCG Gene Signatures

We further characterised the gene signatures upregulated by EGCG (23, 11 + 12 genes) and downregulated by EGCG (50, 13 + 37 genes) via the R package EnrichR [80]. Figure 3A,B show that the 50 genes downregulated by the EGCG gene signature are overrepresented in the kidney-associated terms “microalbumin in urine” from the UK_Biobank_GWAS_v1 dataset collection and “kidney cancer” from the “Jensen Diseases” dataset collection.

The enrichment analysis revealed that most EGCG-downregulated genes are associated with transcription factor E2F1 (Figure 3C), while most EGCG-upregulated genes are associated with transcription factors JUN and NFKB1 (Figure 3D). The protein interaction network of these transcription factors expanded by five interacting proteins was generated by STRING-DB (Figure 3E) and could be characterised by the Reactome (Figure 3F) and WikiPathway (Figure 3G) pathways related to neuroinflammation and other inflammatory processes.

### 5.6. Transfer of EGCG-Associated Gene Signatures to Kidney Datasets

We followed up our analysis to investigate to what extent our EGCG-associated gene signatures are expressed in kidney transcriptome data. First, we used the dataset GSE198890, in which HEK (human embryonic kidney) cells were treated with the compound VT01454 derived from marine natural products. In the associated publication, Li et al. reported activation of the Hippo pathway by VT01454 via phosphatidylinositol transfer proteins α and β (PITPα/β) as the direct molecular targets [84]. The overlap with our EGCG signature (Figure 4A) consisted only of the genes GRIK2 (upregulated by VT01454, downregulated by EGCG) and SOX15 (upregulated by VT01454, upregulated by EGCG). Second, we used the dataset GSE159656 containing transcriptome data of mouse kidneys treated with the mushroom Ganoderma lucidum. Romero-Córdoba et al., the authors of the associated study, reported anti-hypercholesterolemic, hypoglycaemic and antioxidant activities for the compounds derived from the mushroom [85]. The treatment with Ganoderma lucidum resulted in a larger overlap with the EGCG signatures (Figure 4B) of six genes downregulated by EGCG (*SMCO3*, *SLC12A1*, *FHAD1*, *FAT3*, *LOX* and *DOK7*) and two genes upregulated by EGCG (*TTC9* and *PM20D1*). We further subjected the biggest subset of the six genes from the Venn diagram comparison to an enrichment analysis via the R package EnrichR [80] and found several processes related to renal disease (Figure 4C), most significantly “abnormal renal water reabsorption”, which were only each due to one single gene, either *SLC12A1* or *LOX*.

**Table 1 antioxidants-12-01599-t001:** NCBI GEO datasets used in this transcriptome analysis of natural products.

GEO_Search	Accession	Cell Type	Treatment	PMID/DOI
(Camellia) AND “Homo sapiens”	GSE152781	human skin	EGCG (green tea)	PMID: 33917842 [73]
green tea AND kidney	GSE41644	Mouse colon	EGCG (green tea)	PMID: 23643524 [75]
human T cells EGCG	GSE53448	human CD4^+^CD25^−^ T cells	EGCG (green tea)	PMID: 24476360 [74]
kidney AND “natural products”	GSE198890	HEK cells	VT01454	https://doi.org/10.1038/s41589-022-01061-z (accessed on 1 June 2023) [84]
kidney AND “natural products”	GSE159656	mouse kidney	Ganoderma lucidum	PMID: 33374283 [85]

### 5.7. Comparison of Ganoderma Lucidum-Associated Gene Signatures to Datasets of Kidney Disease Models

As mentioned above, we investigated kidney organoids treated with nephrotoxin puromycin aminonucleoside (PAN) and urine-derived podocytes treated with angiotensin II (ANG II) in two previous publications [82,83]. We compared the GL-treated mouse kidney cell-derived gene signatures with the PAN and ANGII (Figure 5B) gene signatures and found the biggest subset of 100 genes overlapping between GL-upregulated and PAN-downregulated (Figure 5A), 36 genes between GL-downregulated and PAN-downregulated (Figure 5A), 90 genes between GL-upregulated and ANGII-downregulated (Figure 5B) and 76 genes between GL-upregulated and ANGII-upregulated (Figure 5B). These subsets were further explored for overrepresented GO terms, of which the dot plots in Figure 5C–F show the most significant terms. In the 100 genes upregulated in GL and downregulated in PAN, these include VEGF-signalling (Figure 5C,D). The 36 genes downregulated in GL and downregulated in PAN are associated with GO terms related to sterol synthesis and blood circulation/pressure (Figure 5D). The most significant GO terms in the 90 genes upregulated in GL and downregulated in ANGII include epithelial cell development, inflammatory processes such as interferon-alpha and leukaemia inhibitory factor responses and ERBB and BDNF signalling (Figure 5E) and, in the 76 genes upregulated in GL and upregulated in ANGII, include ADP synthesis and metabolism, several other metabolic processes and the positive regulation of acute inflammatory processes (Figure 5F).

Taking into account the inflammatory processes playing major roles in the development of CKD and AKI, we focussed on GOs associated with inflammation and wound responses and found an overlap of the GL treatment with the ANGII (Table 2) but not with the PAN treatment. The genes upregulated in the GL treatment and upregulated in the ANGII treatment (*C3* and *OSMR*) are associated with the “positive regulation of the acute inflammatory response” (*p* = 0.0044), and the genes upregulated in the GL treatment and downregulated in the ANGII treatment (*DDR1*, *DSP*, *HMOX1*, *ID3*, *PLAT*, *PLEC*, *USF1*, *ZFP36* and *ZFP36L2*) are associated with “response to wounding” (*p* = 0.0207).

## 6. Discussion

In this meta-analysis, we first determined the gene signatures in human skin and T cells and mouse colons up- and downregulated by treatment with EGCG, the active ingredient in green tea. The anticarcinogenic, antioxidative and anti-inflammatory properties of EGCG have been shown in a large body of literature [86,87,88]. The large overlap we found between the GO terms associated with EGCG signatures from different cell types in men and mice indicates the conservation of these biological processes. The EGCG-upregulated signatures can be classified by GO terms related to metabolism (amine and phenol metabolism), ion transport, membranes and secretion. The EGCG-downregulated signatures can be classified by GO terms related to calcium signalling, neuronal and synaptic, inflammatory (response to wounding), connective tissue, ion transport and taxis and—concerning the kidneys—with microalbumin in urine and renal cancer. Calcium signalling and the influx and overload of cytosol and mitochondria, have been implicated in oxidative stress and the subsequent development of diseases [89,90]. Among the genes associated with the “response to wounding”, we found neuregulin-1 (*NRG1*) overlapping with downregulated genes in PAN-treated kidney organoids. NRG1 has been reported to attenuate the development of nephropathy in a diabetes I mouse model [91].

As significantly enriched transcription factors E2F1, NFKB1 and JUN were identified and furthermore found at the core of a gene regulatory network controlling inflammatory processes, E2F1 and NFKB1 have been shown to mediate inflammatory processes [92,93] and are connected to JUN/JNK [94,95]. That the same mechanisms are active in renal cells was shown by Liang et al., who reported that EGCG reduces kidney cell inflammation through the NF-κB pathway in HEK cells [96].

We compared this signature to transcriptomes of kidney cells treated with other natural antioxidants. The comparison to HEK cells treated with the marine natural product-derived compound VT01454 identified only *GRIK2* and *SOX15*. The comparison to mouse kidneys treated with the mushroom Ganoderma lucidum resulted in a bigger overlap of six genes downregulated by EGCG (*SMCO3*, *SLC12A1*, *FHAD1*, *FAT3*, *LOX* and *DOK7*) and two genes upregulated by EGCG (*TTC9* and *PM20D1*). Genes downregulated by EGCG, especially *SLC12A1* or *LOX*, can be associated with several processes related to renal disease, most significantly “abnormal renal water reabsorption”.

We compared the Ganoderma lucidum-associated gene signatures to datasets of kidney disease models: (i) kidney organoids treated with the nephrotoxin PAN [82] and (ii) urine-derived podocytes treated with ANG II [83]. The largest intersection subset found in the 100 genes upregulated in GL and downregulated in PAN points at the therapeutic effects of GL on damage induced by PAN as a model for kidney disease. A GO term associated with these 100 genes was VEGF signalling, which we also found to be a major player in CKD in previous publications [4,97,98]. Also, in comparison to ANGII treatment, the largest subset was found in the 90 genes upregulated in GL and downregulated in ANGII. These 90 genes are associated with GO terms related to epithelial cell development, inflammatory processes such as interferon-alpha and leukaemia inhibitory factor responses and ERBB and BDNF signalling, while the 76 genes upregulated in both GL and in ANGII are associated with ADP synthesis and metabolism, several other metabolic processes and the positive regulation of acute inflammatory processes. Following up on these inflammatory processes in more detail, we identified the genes *C3* and *OSMR* upregulated in the GL and ANGII stimulations, associated with the “positive regulation of the acute inflammatory response” (*p* = 0.0044), and the genes *DDR1*, *DSP*, *HMOX1*, *ID3*, *PLAT*, *PLEC*, *USF1*, *ZFP36* and *ZFP36L2* upregulated in the GL treatment and downregulated in the ANGII treatment, associated with the “response to wounding” (*p* = 0.0207).

## 7. Conclusions

Plants with antioxidative and anti-inflammatory properties are abundant in nature and can be found globally. The studies so far have shown that plant-based extracts contain various phytoconstituents that give them their respective properties. Many cases of AKI are too serious to be able to manage them directly using natural products. However, they may be useful in reducing the risk of AKI and in the exploration of disease-associated mechanisms and drug targets for AKI and CKD.

Our meta-analysis of transcriptome data and comparisons to in vitro kidney disease models has demonstrated that EGCG and GL have great potential for therapeutic effects mitigating inflammatory processes in kidney diseases. This has partially already been shown for EGCG in other in vitro models, e.g., by Kanlaya et al. [88]; this, together with our findings, suggests that further studies evaluating the therapeutic effects of GL and EGCG in models of human kidney disease might be fruitful. There are numerous unexploited plant species and herbs that might have antioxidant and anti-inflammatory activities; hence, we recommend the exploitation, and detailed studies on the mechanisms and targets, of these natural antioxidants and anti-inflammatories and the potential use of these drug targets for the management of kidney-associated diseases such as AKI and CKD.

## Figures and Tables

**Figure 1 antioxidants-12-01599-f001:**
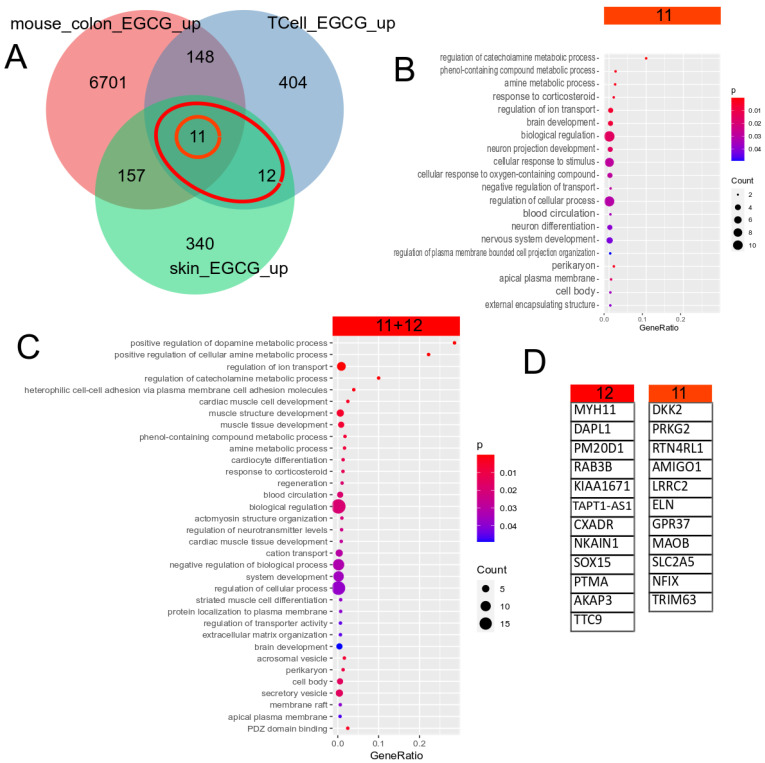
Gene signature upregulated by EGCG. (**A**) Venn diagram comparison of genes upregulated in EGCG-treated human skin cells, human T cells and mouse colon cells. The orange circle marks the 11 genes upregulated in common in all 3 datasets, and the red ellipse marks the 23 (11 + 12) genes upregulated in common in the two human datasets. (**B**) Overrepresented GO terms in the common signatures of the 11 genes. (**C**) Overrepresented GO terms in the common human EGCG signatures of the 23 genes. The large overlap between both GO analysis results shows that most of these biological processes are conserved between men and mice. The results can be grouped into metabolic (amine and phenol metabolism), ion transport, membrane and secretory categories. (**D**) The EGCG gene signatures from the marked subsets of the Venn diagram.

**Figure 2 antioxidants-12-01599-f002:**
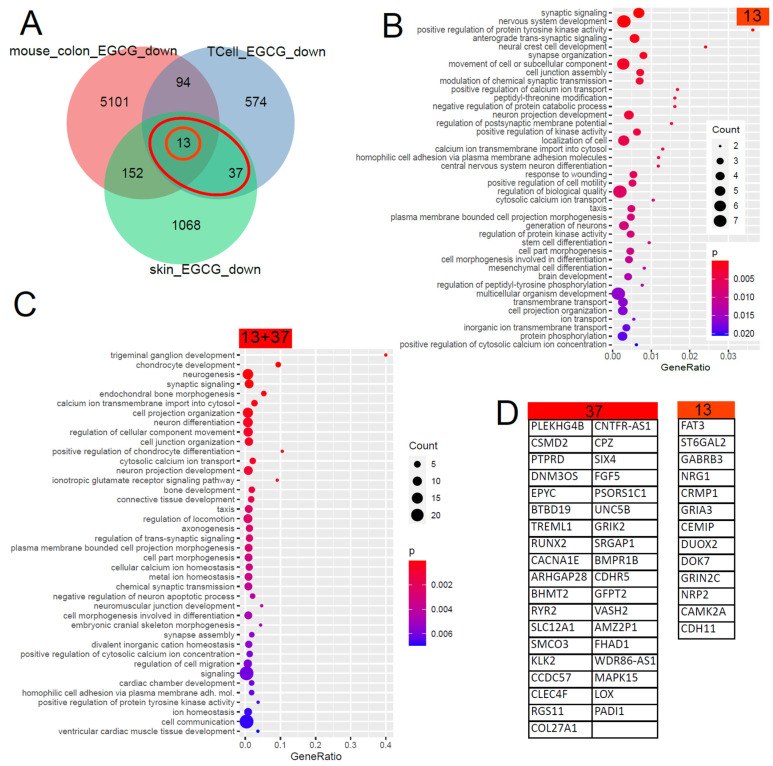
Gene signature downregulated by EGCG. (**A**) Venn diagram comparison of the genes downregulated in EGCG-treated human skin cells, human T cells and mouse colon cells. The orange circle marks the 13 genes downregulated in common in all 3 datasets, and the red ellipse marks the 50 (13 + 37) genes downregulated in common in the two human datasets. (**B**) Overrepresented GO terms in the common signatures of the 13 genes. (**C**) Overrepresented GO terms in the common human EGCG signatures of the 50 genes. The large overlap between both GO analysis results shows that most of these biological processes are conserved between men and mice. The results can be grouped into calcium signalling, neuronal and synaptic, inflammatory (response to wounding), connective tissue, ion transport and taxis categories. (**D**) The EGCG gene signatures from the marked subsets of the Venn diagram.

**Figure 3 antioxidants-12-01599-f003:**
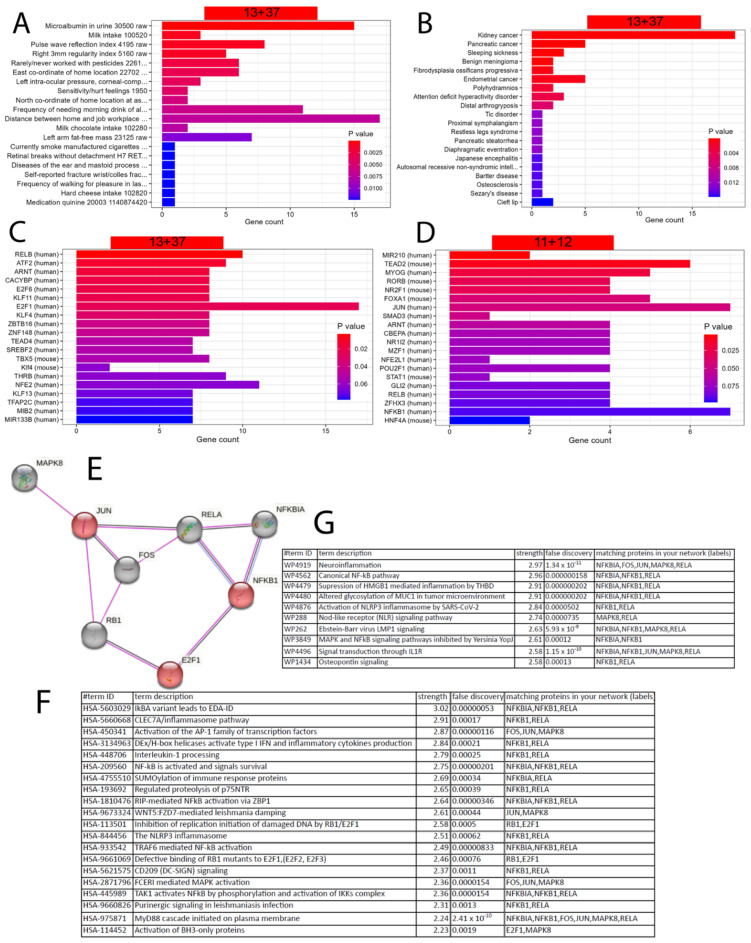
EGCG-associated gene signatures may have an impact on the kidneys and are associated with inflammatory processes regulated by the transcription factors NFKB1, JUN and E2F1. (**A**) The enrichment analysis results of the EGCG-downregulated 50 (13 + 37) genes in the UK_Biobank_GWAS_v1 dataset collection and (**B**) the “Jensen Diseases” dataset collection. (**C**) Most EGCG-downregulated genes are regulated by transcription factor E2F1. (**D**) Most EGCG-upregulated genes are regulated by transcription factors JUN and NFKB1. (**E**) Protein interaction network of E2F1, NFKB1 and JUN expanded by five interacting proteins made via STRING-DB. (**F**) Top 20 Reactome pathways found by STRING-SB in the network of (**E**) reveal inflammatory processes. (**G**) Top 20 WikiPathway pathways found by STRING-SB in the network of (**E**) neuroinflammation and other inflammatory processes.

**Figure 4 antioxidants-12-01599-f004:**
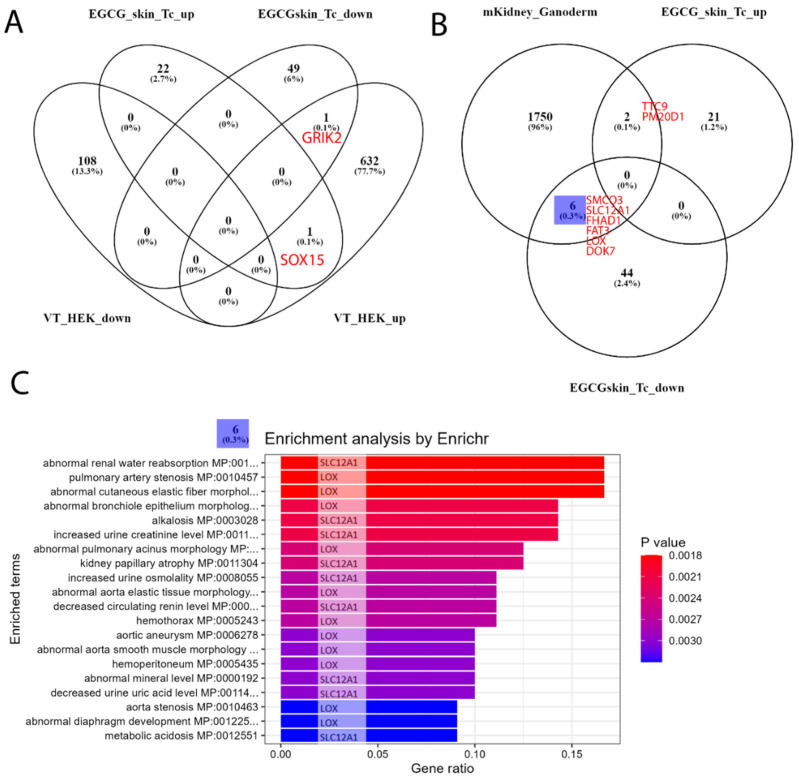
EGCG-associated gene signatures are transferred to kidney datasets with other antioxidant compounds. (**A**) Venn diagram comparison of the human skin and T-cell up- and downregulated EGCG gene signatures to dataset GSE198890 of HEK cells treated with VT01454 shows only 1+1 genes overlapping. (**B**) In dataset GSE159656 of Galoderma lucidum-treated mouse kidney cells, there is more overlap of the indicated six genes in the downregulated EGCG signature and two genes in the upregulated EGCG signature. (**C**) The six EGCG-downregulated genes overlapping can be further characterised by the overrepresented renal impairment processes found by the R package EnrichR; however, they are only associated with single genes: SLC12A1 or LOX.

**Figure 5 antioxidants-12-01599-f005:**
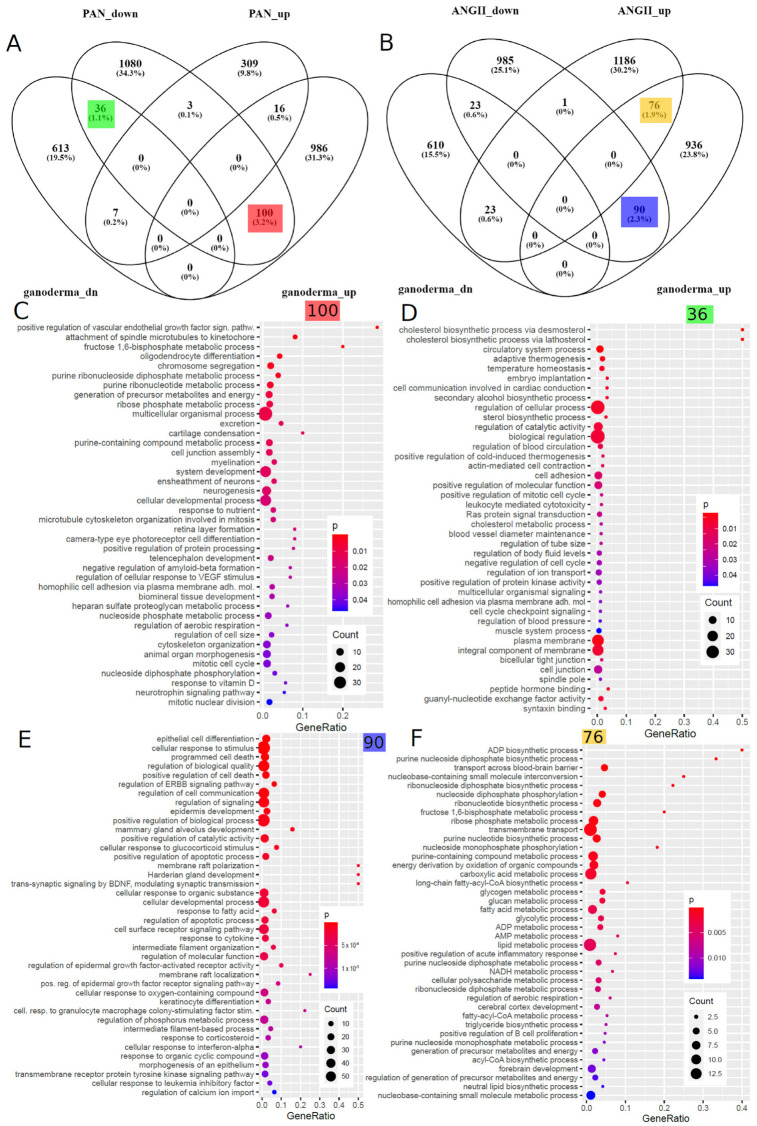
Ganoderma lucidum-associated gene signatures are compared to kidney datasets with PAN and ANGII stimulation. (**A**) Venn diagram comparison of up- and downregulated genes in Ganoderma lucidum (GL)-treated mouse kidney and PAN-treated human kidney organoids. (**B**) Venn diagram comparison of up- and downregulated genes in GL-treated mouse kidney and ANGII-treated human urine-derived podocytes. The dot plots in (**C**–**F**) show the most significant overrepresented GO terms in the Venn diagram subsets with matching colours. (**C**) The most significant GO terms in the 100 genes upregulated in GL and downregulated in PAN include VEGF signalling. (**D**) The most significant GO terms in the 36 genes downregulated in GL and downregulated in PAN include sterol synthesis and blood circulation/pressure. (**E**) The most significant GO terms in the 90 genes upregulated in GL and downregulated in ANGII include epithelial cell development, inflammatory processes such as interferon-alpha and leukaemia inhibitory factor responses and ERBB and BDNF signalling. (**F**) The most significant GO terms in the 76 genes upregulated in GL and upregulated in ANGII include ADP synthesis and metabolism and several other metabolic processes, as well as the positive regulation of acute inflammatory processes.

**Table 2 antioxidants-12-01599-t002:** Inflammatory and wounding-related GO terms overrepresented in the genes overlapping between the GL and ANGII signatures.

Term	Ganoderma Lucidum	ANGII	*p*-Value	Odds Ratio	Genes
positive regulation of acute inflammatory response	up	up	0.0043	22.34	*C3*, *OSMR*
response to wounding	up	down	0.0207	5.41	*DDR1*, *DSP*, *HMOX1*, *ID*, *PLAT*, *PLEC*, *USF1*, *ZFP36*, *ZFP36L2*
wound healing	up	down	0.0473	2.65	*DDR1*, *DSP*, *HMOX1*, *PLAT*, *PLEC*, *USF1*

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
