# Peer review of "Natural Products in Renal-Associated Drug Discovery"

_antioxidants, 2023, doi:10.3390/antiox12081599_

Round 1

Reviewer 1 Report (Previous Reviewer 2)

Expanding on the information would have made the article more complete but the authors mention a limitation for size or words. They have addressed the issues otherwise.

Reviewer 2 Report (Previous Reviewer 1)

I think this manuscript will be acceptable.

This manuscript is a resubmission of an earlier submission. The following is a list of the peer review reports and author responses from that submission.

Round 1

Reviewer 1 Report

Plants in nature with anti-inflammatory and antioxidant effects are widely distributed and used all over the world. This is a review that explores the active  ingredients of plants which were traditionally known to  have those effects.

Although ‘1’. is called Introduction, there are no chapter number after that. Considerable revision as below and more  chapters  between “Introduction” and “Discussion” are needed.

Major

If the introduction is continued to the meta-analysis of P4, I read them with much interest. However,  I have the concern whether detailed description of each plant  in pages 3 and the first half of 4 is necessary to introduce the meta-analysis of transcription. 

It is difficult to agree that it might be useful to prevent AKI. The cause of AKI may be too serious to prevent by natural products especially in humans.

Minor: The characters in figures are too small font size to read manuscript as A4 format.

Reviewer 2 Report

This is a well written review article focussing on natural products and potential to reduce renal disease. 

The concern is that the natural products are listed with very little information. the manuscript should expand on the information known especially for cellular  changes, signaling and potential active components. 

Pgs 3-4 are just lists of compounds and potential activity. This section should be expanded as there is no mention of cellular mechanisms, if the plant was a water or ethanol  extract and which part of the plant was isolated. The manuscript is just too much of an overview. 

It would be beneficial if the authors expand on the conditions for the effects of ECGC on skin and how this relates to HEK cells and renal disease. This appears to digress form the main focus of the manuscript.